# Neuroprotective Potential of Verbascoside Isolated from *Acanthus mollis* L. Leaves through Its Enzymatic Inhibition and Free Radical Scavenging Ability

**DOI:** 10.3390/antiox9121207

**Published:** 2020-11-30

**Authors:** Carmen Burgos, Dolores Muñoz-Mingarro, Inmaculada Navarro, Carmen Martín-Cordero, Nuria Acero

**Affiliations:** 1Department of Pharmacology, Faculty of Pharmacy, Seville University, C/P. García González s/n, 41012 Seville, Spain; carburmar@hotmail.com (C.B.); carmenmc@us.es (C.-M.C.); 2Chemistry and Biochemistry Department, San Pablo-CEU University, CEU Universities, Urb. Montepríncipe, 28668 Boadilla del Monte, Madrid, Spain; dmumin@ceu.es; 3Department of Physical Chemistry, Faculty of Pharmacy, Seville University, C/P. García González s/n, 41012 Seville, Spain; inzafra@us.es; 4Pharmaceutical and Health Sciences Department, San Pablo-CEU University, CEU Universities, Urb. Montepríncipe, 28668 Boadilla del Monte, Madrid, Spain

**Keywords:** *Acanthus mollis* L., verbascoside, neuroprotection, antioxidant, natural compounds, oxidative stress, acteoside

## Abstract

The phenomenon of today’s ageing population has increased interest in the search for new active substances that delay the onset and development of neurodegenerative diseases. In this respect, the search for natural compounds, mainly phenolic compounds, with neuroprotective activity has become the focus of growing interest. Verbascoside is a phenylethanoid that has already presented several pharmacological activities. The purpose of this study is to isolate and identify verbascoside from *Acanthus mollis* leaves. Consequently, its neuroprotective ability through enzymatic inhibition and free radical scavenging ability has been analyzed both in vitro and in cell culture assays. The antioxidant capacity of verbascoside was evaluated in vitro through total antioxidant capacity, DPPH^•^, ^•^OH, and O_2_^•^—scavenging activity assays. The effect of verbascoside on intracellular reactive oxygen species (ROS) levels of HepG2 and SH-SY5Y cell lines was studied in normal culture and under induced oxidative stress. The inhibitory ability of the phenylethanoid against several enzymes implied in neurodegenerative diseases (tyrosinase, MAO-A, and AChE) was analyzed in vitro. Verbascoside neuroprotective activity is at least in part related to its free radical scavenging ability. The effect of verbascoside on ROS production suggests its potential in the prevention of harmful cell redox changes and in boosting neuroprotection.

## 1. Introduction

In the last 100 years, human life expectancy has strikingly increased, and with it the average age of the population. This progressive ageing of society is associated with a greater prevalence of age-related illnesses, which constitutes a major problem for public health. A large proportion of people over the age of 60 suffer from cognitive decline, memory deficits, and changes in brain structure [1]. Alzheimer’s disease (AD) and Parkinson´s disease (PD) have become the most common neurodegenerative disorders. In this respect, oxidative stress is strongly implied in the progression of neuronal degeneration [2].

The enhancement of cholinergic transmission has been considered one of the main therapeutic objectives for AD treatment [3]. Acetylcholinesterase (AChE) inhibitors can mitigate cholinergic deficits and increase neurotransmission by raising acetylcholine levels, which results in an improvement in the cognitive function of these patients [4].

Tyrosinase catalyzes tyrosine oxidation, which constitutes a limiting stage of melanin synthesis. Therefore, this enzyme plays a fundamental role in human skin, eye, and hair pigmentation. Furthermore, a relationship between melanoma, PD, and this enzyme also exists. Melanin-related alterations are associated with skin cancer occurrence, such as melanoma, while those alterations in neuromelanin are correlated with neurodegenerative diseases such as PD. This fact suggests that melanin, and thus tyrosinase, could be significant factors in the alteration of vulnerability to these ailments [5], although the precise mechanisms of their involvement have yet to be elucidated [6]. 

Both monoamine oxidase (MAO) isoforms, A and B, catabolize monoamine neurotransmitters, and therefore constitute therapeutic targets for neuropsychiatric and neurodegenerative diseases [7]. MAO-A inhibitors increase the levels of noradrenaline and serotonin in the brain and are employed to treat anxiety and depression. On the other hand, MAO-B inhibitors are useful for PD symptomatic therapy [8].

Another key factor in neurotoxicity and cell death is oxidative stress, which is responsible, at least in part, for the onset and development of neurodegenerative disorders [9]. Oxidative stress is directly related to the production and rising levels of reactive oxygen species (ROS), reactive nitrogen species, and superoxide radicals [10]. Free radicals cause oxidative damage in essential cellular biomolecules, such as proteins, lipids, and nucleic acids, by altering their structure and affecting their function [11]. Consequently, they could trigger cell apoptosis processes [12]. During cellular ageing, the amount of these radicals increases, and therefore a high ROS production supposes a risk factor in the pathogenesis and progression of many diseases [13,14].

Verbascoside (also known as acteoside and kusaginin) is a phenylethanoid glycoside: 2-(3,4-dihydroxyphenyl)ethyl-1-Ο-α-L-rhamnopyranosyl-(1→3)-(4-Ο-Ε-caffeoyl)-β-D-glucopyranoside (Figure 1). This phenolic compound is a product of the shikimic acid pathway, but the synthesis and enzymes involved therein have yet to be completely elucidated [15].

Nowadays there is a growing interest in phenolic compound studies due to their potential health benefits, such as in the prevention of cancer and cardiovascular diseases, and even of neurodegenerative disorders such as AD. In this respect, plants with high amounts of verbascoside have traditionally been used for a variety of purposes, which aroused interest in this compound [16].

Several beneficial effects have been reported regarding this compound, including antibacterial, cytoprotective, antitumoral, wound-healing, antioxidant, gastroprotective, neuroprotective, and photoprotective effects, among others [17].

Verbascoside blocks amyloid deposition and reduces amyloid *β* peptide toxicity. This phenylethanoid inhibits amyloid *β* peptide oligomerization and enhances its degradation. As a result of these activities, acteoside appears as a promising therapeutic agent for AD [18]. Neuroprotective effects are also indirectly improved by the antioxidative effects of this molecule. In this respect, it increases the main cellular antioxidant enzymes such as gluthathione peroxidase and reductase, catalase, and superoxide dismutase [19,20].

*Acanthus mollis* L. is a herbaceous plant from the Acanthaceae family. Native to the Mediterranean basin, it has many traditional medicinal uses. Its leaves are used as a diuretic and analgesic for digestive and urinary pain. Their infusion is employed in southern Italy, Porto Santo, and Madeira for the treatment of psoriasis, atopic dermatitis, other irritative skin disorders, tumors, ulcers, and as an anti-inflammatory for swollen legs. In Sardinia, Spain, and Portugal, the leaves are boiled and applied in a poultice as an emollient or as skin protection. They are also recommended for the treatment of wounds and burns, hemorrhoids, toothache, and inflammation of the oral cavity [21]. To date, several pharmacological studies have reported the anti-inflammatory and anti-hemorrhoidal capacity of *A. mollis* leaves, as well as the antifungal and antioxidant capacity of certain extracts [22]

The purpose of this study was to analyze neuroprotective and free radical scavenging capacity through in vitro and in cell culture assays of verbascoside isolated from *A. mollis* leaves. Each antioxidant evaluation differs in terms of the radicals generated, the analytical strategies, the antioxidant capacity evaluated, and the assay sensitivity. Thus, antioxidation needs to be evaluated by a variety of assays in order to understand the molecule mechanism of action. In this study, the DPPH^•^ and ^•^OH radical scavenging capacity was analyzed, as were the effect of verbascoside on the xanthine/xanthine oxidase system (studying both the effect against the enzyme and the superoxide scavenging activity) and the total antioxidant activity. The in vitro neuroprotective potential was evaluated through the analysis of the effect of the phenylethanoid against several enzymes implied in neurodegenerative diseases: tyrosinase, MAO-A, and AChE. For cell culture studies, two cellular lines were used: HepG2 as a human hepatocyte model [23] and SH-SY5Y to evaluate neuroprotection [24]. The effect of verbascoside on the intracellular ROS levels of those cells was evaluated in normal culture and under induced oxidative stress.

## 2. Materials and Methods

### 2.1. General Experimental Procedure and Chemicals

Nuclear magnetic resonance (NMR) spectra were recorded in dimethyl sulphoxide (DMSO)-d6 on a Bruker Avance 500 spectrometer operating at 500 MHz (1H) and 125 MHz (13C). Chemical shifts (δ, in ppm) were referenced to tetramethylsilane (TMS), and J was expressed in Hz. FAB-MS was taken on a Thermo Scientific TSQ 8000 Evo Triple Quadrupole mass (70eV) spectrometer (Thermo Fisher Scientific, USA). The UV-Vis spectrum was recorded in methanol on a Shimadzu UV-1800 spectrophotometer with silica gel column chromatography (Merck, 107734, Darmstadt, Germany). Ethanol, methanol, petroleum ether, ethyl acetate, and n-butanol were obtained from Panreac^®^ (Barcelona, Spain). All chemicals were of analytical reagent grade. A quantity of 2,2-diphenyl-1-picrylhydrazyl (DPPH) in free radical form and acetylcholinesterase (AChE) was purchased from Sigma (Merck KGaA, Darmstadt, Germany).

### 2.2. Plant Material and Extract Preparation

*Acanthus mollis* L. (Acanthaceae) was collected in September 2016 in Aravaca (Madrid, Spain) (40º 27´13.8” N, –3º 45´37.5” W). A voucher specimen (Ref. 3372) was deposited in the Faculty of Pharmacy Herbarium, University San Pablo, CEU Madrid. The leaves (300 g) of this plant were dried at room temperature, powdered, and extracted three times with 3 L of ethanol (EtOH) in an ultrasonic bath (Ultrasons, Selecta^®^, Barcelona, Spain) at 60 °C for 1 h. The ethanolic extract was evaporated to dryness (40.11 g), suspended in 50 mL of distilled water, and then extracted successively with petroleum ether (PET), ethyl acetate (EtOAc), and n-butanol (n-BuOH).

### 2.3. Extract Fractionation and Isolation

The ethyl acetate extract (4 g) was subjected to silica gel column chromatography with EtOAc:MeOH:H_2_O (80:3:3) to give compound **1** (80 mg) as white needles after recrystallization with MeOH. This compound was characterized by spectroscopic analyses, including ^1^H NMR, ^13^C NMR, 2D NMR (^1^H-^1^H COSY, HMBC), FAB-MS, and UV.

### 2.4. Cell Culture

The HepG2 human hepatocarcinoma and the human neuroblastoma SH-SY5Y cell lines were obtained from the European Collection of Cell Cultures (Health Protection Agency, London, UK) (ECACC Ref.-85011430 and Ref.-94030304, respectively). The HepG2 cell line is a model that reproduces human hepatocyte and has been widely used in the evaluation of the antioxidant effects of various natural compounds [23]. On the other hand, SY-SY5Y is commonly used as an in vitro model of neural function and neural differentiation [25]. HepG2 cells were grown in EMEM (Eaglee’s Minimum Essential Medium) and SH-SY5Y in Ham’s F-12: EMEM (EBSS) 1:1, supplemented both with 2 mM glutamine, 1% non-essential amino acids, inactivated fetal bovine serum (FBS) (HyClone, Logan, UT, USA) (10% for HepG2 and 15% for SH-SY5Y), and 1% antibiotics (10,000 units of penicillin and 10 mg/mL of streptomycin). Both HEPG2 and SY-SY5Y were cultured at 37 °C in a humidified 5% CO_2_ atmosphere. For antioxidant assays, cells were incubated in a medium supplemented with only 1% FBS in order to prevent cytotoxic artefacts from forming as a result of the possible interaction between FBS components and phenolic compounds [26]. The chemicals were obtained from Sigma (Merck KGaA, Darmstadt, Germany).

### 2.5. In Vitro Antioxidant Activity Assay

#### 2.5.1. DPPH^•^ Scavenging Assay

The 2,2-diphenyl-*β*-picrylhydrazyl (DPPH) assay was performed to test the in vitro free radical scavenging activity of verbascoside. This assay consists of analyzing the capacity of the verbascoside to reduce the stable radical DPPH^•^ by donating a hydrogen, thereby forming DPPH-H. This reduction results in a color change in the DPPH, which goes from purple (oxidized form) (absorption band 517 nm) to yellow (reduced form) [27]. In a 96-well plate, 100 µL of DPPH 1 mM in MetOH and 100 µL of verbascoside or ascorbic acid were mixed at various concentrations in MetOH. The plate was incubated for 20 min at room temperature in darkness, and the absorbance was measured in a Versa Max (Molecular Devices) plate reader at 517 nm.

The percentage of DPPH reduction was calculated by using the formula:Reduction % = [(A_0_ − A_1_)/A_0_] × 100(1)
where A_0_ is the absorbance of the control and A_1_ is the absorbance of verbascoside or ascorbic acid in each concentration. With this data, the IC_50_ for each compound was determined as the concentration capable of reducing the DPPH radical by 50%.

The results were expressed as the mean IC_50_ of the three replicates.

#### 2.5.2. Xanthine/Xanthine Oxidase Assay

The xanthine/xanthine oxidase system generates superoxide radicals with the capacity to reduce NBT (nitro blue tetrazolium). Antioxidant substances can capture such radicals and prevent the reduction of NBT. Verbascoside capture capacity at various concentrations was quantified by measuring the absorbance changes at 560 nm over time [28]. Ascorbic acid was used as the reference compound.

All reagents were prepared in a phosphate buffer (50 mM of KH_2_PO_4_/KOH, pH 7.4). For blank preparation, 100 µL of phosphate buffer, 10 µL of EDTA 15 mM, 15 µL of hypoxanthine 3 mM, and 25 µL of NBT 0.6 mM were mixed. For positive control, 75 µL of phosphate buffer 50 mM, 10 µL of EDTA 15 mM, 15 µL of hypoxanthine 3 mM, 25µL of NBT 0.6 mM, and finally, 25 µL of xanthine oxidase (1 unit per 10 mL of buffer) were added. Finally, in order to evaluate the activity of verbascoside or the reference compound, in each well, 62.5 µL of buffer phosphate, 10 µL of EDTA 15 mM, 15 µL of hypoxanthine 3mM, 12.5 µL of verbascoside or ascorbic acid at different concentrations, 25 L µL of NBT 0.6 mM, and 25 µL of xanthine oxidase (1 unit/10 mL of buffer) were mixed.

Absorbance measurements were performed for 40 min at 5-min intervals on the Versa Max (Molecular Devices) plate reader at 560 nm. The results are expressed as a percentage of inhibition of NBT reduction.

In order to analyze the ability of the phenylpropanoid to inhibit the enzyme xanthine oxidase, the amount of uric acid produced (295 nm) was determined, and the results were expressed as a percentage of inhibition of uric acid production. For this purpose, 600 µL of phosphate buffer 50 mM pH 7.4, 60 µL of EDTA 15 mM, 90 µL of xanthine 0.1 mM, 75 µL of buffer or verbascoside at various concentrations, and finally 150 µL of xanthine oxidase (1 unit/10 mL of buffer) were mixed. Absorbance measurements were performed for 40 min at 5-min intervals on the Versa Max (Molecular Devices) plate reader. The results were expressed as a percentage of inhibition of the enzyme xanthine oxidase. All tests were conducted in triplicate.

#### 2.5.3. Hydroxyl Radical Scavenging Capacity

The reaction mixture for the determination of hydroxyl radical scavenging capacity of verbascoside and a reference substance consisted of 1 mL of FeSO_4_ 1.5 mM, 0.7 mL of hydrogen peroxide 6 mM, 0.3 mL of sodium salicylate 20 mM, and 1 mL of different concentrations of verbascoside or ascorbic acid (0.07–0.33 mg/mL). After 1 h of incubation at 37 °C, the hydroxyl complex absorbance was measured at 562 nm on the Asys UVM 340 plate reader.

The scavenging percentage capacity was calculated using the following equation:Scavenging % = [1 − (A_1_ − A_2_)/A_0_] × 100(2)
where A_0_ is the absorbance of the control, A_1_ is the absorbance of verbascoside or of the reference compound, and A_2_ is the absorbance of verbascoside or of the reference compound without sodium salicylate [29]. All tests were conducted in triplicate.

#### 2.5.4. Total Antioxidant Capacity (TAC)

For the determination of total antioxidant capacity, the phosphomolybdene reduction assay was used in accordance with the protocol by Prieto et al. [30]. To this end, 300 µL of different concentrations of ascorbic acid or verbascoside were added to 3 mL of a reagent consisting of ammonium molybdate 4 mM, sodium phosphate 28 mM, and sulphuric acid 0.6 mM. The mixture was kept at 90 °C in a water bath for 90 min; it was then cooled, and absorbance was measured at 695 nm on the Asys UVM 340 plate reader. The results are expressed as ascorbic acid equivalents (µg/mL). All tests were conducted in triplicate.

#### 2.5.5. Cell Culture Antioxidant Capacity. Intracellular ROS Measurement

In order to determine the effect of verbascoside on the concentration of intracellular ROS, and hence to ascertain its antioxidant capacity in cell culture, the 2,7-dichlorodihydrofluorescein diacetate (DCFH-DA) assay was performed [31]. This molecule passes through the cell membrane and is deacetylated inside the cell, thereby forming the 2,7-dichlorodihydrofluorescein (DCFH). DCFH in the presence of ROS is oxidized to 2,7-dichlorofluorescein (DCF) and emits fluorescence. Therefore, the fluorescence detected in the test is directly proportional to the intracellular ROS concentration.

This assay was conducted to analyze the possible antioxidant effect of verbascoside in two situations: (a) measuring its direct effect on cells growing under normal conditions; and (b) pre-incubating cells with different concentrations of verbascoside before inducing oxidative stress with hydrogen peroxide, which we will refer to as the protective effect.

Eight thousand cells per well were seeded in 96-well plates. Plates were incubated for 24 h at 37 °C in a 5% CO_2_ atmosphere.

For the study of the direct effect, after 24 h of incubation, the medium was replaced by 200 µL/well of DCFH-DA 0.02 mM in phosphate-buffered saline (PBS). After 30 min of incubation, cells were washed with PBS, and different concentrations were added of verbascoside dissolved in the corresponding cell culture medium supplemented with 1% FBS. The measurement of fluorescence was initiated once the verbascoside had been added.

For the protective effect, cells were pre-treated for 24 h with different concentrations of verbascoside dissolved in the appropriate culture medium for each line and supplemented with 1% fetal bovine serum. Subsequently, the medium was replaced by DCFH-DA 0.02 mM and plates were incubated for 30 min. The cells were then washed with PBS, and the oxidative stress was induced with H_2_O_2_ 200 M for HepG2, and 500 M for SH-SY5Y in the culture medium. The measurement of fluorescence was initiated once the H_2_O_2_ was added.

Fluorescence measurements were performed on a Fluostar Optima (BMG Labtech) plate reader for 90 min at 15-min intervals at an excitation wavelength of 485 nm and an emission wavelength of 520 nm. The means of the measurements performed at each time interval were calculated for each concentration. The results were expressed as fluorescence versus time, for each treatment.

All trials were conducted in triplicate.

### 2.6. Neuroprotective Capacity. In Vitro Enzymatic Inhibition Capacity

#### 2.6.1. Tyrosinase Inhibition Assay

In order to determine the possible verbascoside effect on enzyme activity, dopachrome production was measured at 475 nm. The reaction was carried out on a 96-well plate. In each well, 10 µL of verbascoside was added at different concentrations (or *α*-Kojic acid, which was used as a reference substance, or 10 µL of H_2_O for control), together with 40 µL of L-DOPA 5 mM in phosphate buffer 63 mM pH 6.8, 80 µL of phosphate buffer 63 mM pH 6.8, and 40 µL of tyrosinase 200 U/mL. All tests were conducted in triplicate.

Absorbance was then measured every 3 min for 15 min on a Spectrostar Nano (BGM Labtech) plate reader at 37 °C [32]. The results are expressed as the percentage of inhibition of the enzyme for each concentration. For the calculation of the inhibitory activity, the slope of the linear change in absorption measured for assays with verbascoside were compared to the slope for blank controls. The IC_50_ (concentration that inhibits 50% of enzyme activity) was calculated for each of the two compounds.

#### 2.6.2. MAO-A Inhibition Assay

Assays were conducted on a 96-well plate as previously described by [33]. In each well, 50 µL of verbascoside at different concentrations dissolved in DMSO; 50 µL of clorgyline, which was used as a reference substance; or 50µL of DMSO as a control were mixed with 50 µL of chromogenic solution (0.8 mM vanillic acid, 0.417 mM of 4-aminoantipyrine, 4 U/mL peroxidase in potassium phosphate buffer 0.2 M pH 7.6), 100 µL of tyramine 3 mM in potassium phosphate buffer 0.2 M pH 7.6, and 50 µL of MAO-A 8 U/mL in potassium phosphate buffer 0.2 M pH 7.6. All tests were conducted in triplicate.

The absorbance was measured every 5 min for 40 min on a Spectrostar Nano (BGM Labtech, Ortenberg, Germany) plate reader at 37 °C. The results were expressed as a percentage of enzyme inhibition for each verbascoside or clorgyline concentration. To this end, the slope of the linear change in absorption measured for assays with verbascoside or clorgyline was compared to the slope for blank controls. The IC_50_ (concentration that inhibits 50% of enzyme activity) was calculated for each of the two compounds.

#### 2.6.3. Acetylcholinesterase (AChE) Inhibition Assay

In order to study the effect of the verbascoside on AChE, a colorimetric assay was conducted in which acetylthiocholine was used as a substrate of the enzyme. The thiol group of thiocholine, produced as a result of the enzymatic hydrolysis of acetylthiocholine, reacts with DTNB (5-5 ditiobis-2-nitrobenzoate), thereby producing the yellow anion 5-thio-2-nitrobenzoic acid, which is quantifiable at 405 nm [34].

The reaction was conducted on 96-well plates. Each well contained 25 µL of verbascoside at different concentrations dissolved in Tris-HCl 50 mM pH 8 A buffer, galantamine, which was used as a positive control, or Tris-HCl buffer 50 mM pH 8 as a control; 25 µL of acetylthiocholine 15 mM (in milli-Q water); 125 µL of DTNB (5,5’-dithiobis-(2-nitrobenzoic acid)) 3 mM in C buffer (Tris-HCl 50 mM pH 8, NaCl 0.1 M, and MgCl_2_ 0.02M); and 50 µL of buffer B (Tris-HCl 50 mM pH 8) with 0.1% BSA (bovine serum albumin)

After 30 s, the absorbance was measured at 405 nm (T0), and 25 µL of acetyl cholinesterase (0.22 U/mL in A buffer) was then added. The absorbance was measured every 90 s for a total of 15 min at 37 °C on a Spectrostar Nano plate reader (BGM Labtech, Ortenberg, German). All tests were conducted in triplicate.

The results were expressed as a percentage of enzyme inhibition for each verbascoside or galantamine concentration. To this end, the slope of the linear change in absorption measured for assays with verbascoside or galantamine were compared to the slope for blank controls. The IC_50_ (concentration that inhibits 50% of enzyme activity) was calculated for each of the two compounds.

### 2.7. Statistical Analysis

ANOVA followed by Bonferroni’s post-hoc comparisons tests were performed in all statistical analyses, using the IBM SPSS Statistics 2.0 program. Statistics with a value of *p* < 0.05 were considered significant and will be indicated in figures with different letters. This means that treatments that do not share a letter were significantly different.

## 3. Results

### 3.1. Isolation and Identification of Compound **1**

The ethanolic extract from *Acanthus mollis* leaves was fractioned with solvents of different polarity. The ethyl acetate extract was separated using silica gel column chromatography and provided compound **1** as white needles after recrystallization with MeOH. This compound was characterized by spectroscopic analyses, including ^1^H NMR, ^13^C NMR, 2D NMR (^1^H-^1^H COSY, HMBC), FAB-MS, and UV, and was identified as verbascoside through comparison of its spectral features with the data available in the literature [35].

### 3.2. In Vitro Antioxidant Activity

The results for the in vitro scavenging capacity of verbascoside and ascorbic acid are summarized in Table 1. The DPPH^•^ radical scavenging assay is widely used to preliminarily test the antioxidant capacity of a plant extract or isolated compounds [36]. The substances that show good results in this assay constitute a reliable basis for future in vitro and in vivo studies. Verbascoside showed great DPPH^•^ scavenging ability (58.1 µM ± 0.6), with better values than those of the reference substance (284.9 µM ± 1.2). Previous studies reported similar DPPH^•^ scavenging values for ascorbic acid (306.7 µM) [37], while others estimated gallic acid IC_50_ in 352.7 µm [38]. Verbascoside with a higher capacity than those reference antioxidant compounds was pointed out as an interesting molecule.

The phenylethanoid under study was able to prevent the NBT reduction during the xanthine/xanthine oxidase assay. This effect could be due to the ability of verbascoside to capture the O_2_^^•^−^ radical, or to its capacity to inhibit the enzyme xanthine oxidase, whereby it produces a lower superoxide radical amount. In order to verify whether acteoside can inhibit this enzyme, uric acid production was studied over time. Concentrations between 2.08 and 83.3 µg/mL were analyzed. After plotting absorbance over time for each concentration, the slopes of the linear part of the obtained graphics were compared with those of the control (three replicates were performed in each case). No differences were found in any of the cases. It can therefore be concluded that this compound, together with ascorbic acid, was able to capture the radical O_2_^^•^−^ with similar efficiency but was unable to inhibit the enzyme xanthine oxidase. This verbascoside capacity could be explained by the action of free hydroxyl groups [36].

Radical superoxide can be transformed into species that are more reactive, such as the hydroxyl radical, and this fact explains its implication in multiple pathophysiological processes [39]. Therefore, superoxide anion is implied in the formation of other ROS, which can induce oxidative damage to lipids, proteins, or DNA, thereby triggering the onset and progression of several diseases [40].

The hydroxyl radical is known to be extremely harmful, as it can initiate self-oxidation, polymerization, and fragmentation of numerous biological molecules [41]. These radicals are the most reactive ROS, and cause damage to essential cell macromolecules, which lead to mutagenesis, carcinogenesis, or simply ageing [42]. Furthermore, hydrogen peroxide can lead to hydroxyl radicals, and hence its elimination is crucial for antioxidant defense in the cellular system [43]. The results obtained for this radical showed that the verbascoside IC_50_ for this radical scavenging capacity is 357 ± 16.8 µM, while for ascorbic acid it is 1031 ± 19.9 µM. The scavenging capacity of hydroxyl radicals appears to be directly related to the prevention of lipid peroxidation [44].

### 3.3. Total Antioxidant Capacity (TAC)

The use of different tests to evaluate the antioxidant activity of a compound does not always lead to easy interpretation. In this respect, the concept of a test that reflects total antioxidant capacity is of interest. The results regarding the TAC are shown in Figure 2.

There is a growing interest in the search for antioxidant compounds from plant sources in order to find pharmacologically potent molecules with little to no side effects. Plants produce many compounds to prevent oxidative stress, and these secondary metabolites present a potential source of substances of pharmacological interest, which could replace synthetic antioxidants [36,45]. Since many diseases, such as those mentioned above, are associated with the overproduction of free radicals (brain ischemia, inflammation, cancer, neurodegeneration, and ageing), antioxidants able to counteract oxidative stress are gaining interest for the prevention and treatment thereof. These compounds are also becoming a popular focus for the food industry [45].

### 3.4. Cell Culture Antioxidant Capacity. Intracellular ROS Measurement

The effect of verbascoside at different concentrations on intracellular ROS levels was evaluated in human hepatocarcinoma and neuroblastoma cell line culture (Figure 3 and Figure 4). After treating the cells with this compound, the fluorescence was quantified, which was proportional to ROS levels in the cell. HepG2 has been established as an in vivo cellular model, where the effect of a potential chemopreventive compound can be reliably tested with minimal variations [23], while SH-SY5Y is frequently used as a neuronal cell model [46].

The effect of verbascoside on cells growing under normal conditions is shown in Figure 3. This is considered the direct effect of the phenylethanoid. In both cell lines, after 90 min of treatment, this compound significantly increased the fluorescence with respect to the control. However, there are differences between the two cell lines. Regarding HepG2, an increase in respect to the control in ROS levels was observed, which was significantly higher in cells treated with the lowest concentration of verbascoside, and less evident with the higher concentrations tested. These results agree with those reported by [47], who observed a greater increase after treatment with the lowest assayed concentrations of verbascoside in the ROS levels for lamb oocytes. On the other hand, in SH-SY5Y, the effect, also dose dependent, contrasts with the previous results, since the highest fluorescence is detected with the highest verbascoside levels.

These results point out that in cell culture after short exposures and under the tested conditions, verbascoside promoted oxidative stress. This cell stress effect could be due to the degradation of the phenylethanoid and the subsequent generation of H_2_O_2_, as has already been described for the oxidation of other phenolic compounds in the culture medium [48].

The cytotoxic effect of ROS has been extensively documented for various types of cells, including neurons, whereby it causes cell death in brain trauma, ischemia, and neurodegenerative diseases [49]. However, ROS can also act as molecular signals in physiological processes such as proliferation, migration, and cell survival [50,51,52]. While the deleterious effects of oxidative stress caused by ROS are very well documented, the beneficial consequences of these molecules on central nervous system neurons remain underestimated [53]. In this respect, ROS can act as neuroprotective by activating tyrosine kinase receptors involved in neural survival and their differentiation, and in synaptic structure, function, and plasticity [54]. Accordingly, the neuroprotective effect of moderate exercise is partially due to the activation of cellular metabolism associated with the production of ROS, which alters cerebral redox signals [13]. Therefore, this small increase in ROS levels caused by verbascoside could imply protective effects for neurons and for hepatocytes, although a more in-depth study is required before any conclusions can be drawn.

After 90 min of the oxidative stress induction with hydrogen peroxide, a significant increase in intracellular ROS levels was observed (stressed control (red line) versus non-stressed control (yellow line) in both cell lines) (Figure 4). Cells that were pre-treated with verbascoside were able to significantly reduce ROS levels with respect to stressed control, although they failed to reduce them to non-stressed control values. This effect is more pronounced in the HepG2 cell line. These results agree with those of Chiaino et al. [55], who recently described the neuroprotective properties of a nutraceutical product, rich in verbascoside from *Olea europaea* L. and *Hibiscus sabdaria* L., toward oxidative stress-mediated injury in human neuroblastoma SH-SY5Y.

Phenols can regulate the expression of antioxidant enzymes through the activation of protein kinase C. This kinase catalyzes Nrf2 phosphorylation and thus its translocation to the nucleus where this molecule activates the expression of antioxidant enzymes [56]. However, verbascoside has been described as a potent protein kinase C inhibitor [57], hence the antioxidant effect of this molecule should not be related with this pathway.

Oxidative stress, which leads to high levels of intracellular ROS, is harmful for cells. In this respect, the ROS scavenging capacity of verbascoside contributes to apoptosis inhibition both in vitro and in vivo [58]. The capacity of this molecule to scavenge free radicals could also be related to its chemopreventive capacity in cell culture under oxidative stress, as shown in this study. Similar effects have been described by [59] in fibroblasts after causing oxidative stress with UV rays. The results obtained herein support the hypothesis of the consumption of phenols as an alternative, or at least as a complementary, therapy for the prevention and treatment of degenerative diseases such as Alzheimer’s [60]. The effect of verbascoside on ROS production reveals its potential not only to prevent harmful cell redox changes, but also to promote neuroprotection.

These results suggest that verbascoside neuroprotective activity is at least in part related to its free radical scavenging ability. This relationship has already been demonstrated for other verbascoside activities, such as its anti-inflammatory or UV-protective capacity [17].

### 3.5. Neuroprotective Capacity. In Vitro Enzymatic Inhibition Capacity

#### 3.5.1. Tyrosinase Inhibition Assay

This is a colorimetric assay that determines the production of dopachrome due to tyrosinase enzyme activity. *α*-Kojic acid was used as the reference substance; the IC_50_ value for this enzyme was 83.7 ± 4.5 µM.

Figure 5 shows the results for verbascoside. This substance, far from inhibiting the tyrosinase enzyme, caused a dose-dependent activation, which is significant despite not being very pronounced even at the highest concentrations. Due to the antibacterial and antifungal effect, both isolated verbascoside and plant extracts with high concentrations of this product (*Camellia sinensis* (L.), Kuntze, or *Commiphora mukul* (Hook. ex Stocks) Engl.), have been postulated as promising substances for the development of pharmacological treatments for acne [61]. Our results discourage this use due to the possibility of the appearance of spots on the skin, since tyrosinase is implied in melanin synthesis.

#### 3.5.2. MAO-A Inhibition Assay

Clorgyline was used as selective inhibitors of MAO-A. The IC_50_ value for this substance, used as a reference inhibitor, was 0.122 ± 0.009 µM. The inhibitory effect of verbascoside is shown in Figure 6.

Verbascoside shows a significant and dose-dependent inhibitory capacity of MAO-A, with an IC_50_ of 3.44 ± 0.06 µM. MAO-A shows a high selectivity by the neurotransmitter serotonin, and hence inhibitors of this enzyme prolong this neurotransmitter action and are useful in the treatment of depression and anxiety [62].

Although MAO-B is the isoform used for the treatment of Parkinson’s disease, most of these patients present signs of depression, and therefore MAO-A inhibitors are also beneficial for these symptoms [63]. In addition, MAO-A inhibitors produce symptomatic benefits by reducing the oxidation of dopamine that is catalyzed by this enzyme. Although there is a higher concentration of MAO-B in the basal ganglia, isoform A inhibitors increase the dopamine concentration to this level. Therefore, inhibition of both MAO-A and B isoforms is more effective by preserving dopamine levels at the basal ganglia than selective inhibition [7]. Oxidative stress in the brain may be decisive in the initiation and progress of neurodegeneration. The ability of verbascoside to inhibit MAO-A and to scavenge free radicals could contribute in a major way towards lowering such stress.

#### 3.5.3. Acetylcholinesterase (AChE) Inhibition Assay

Galantamine is a well-known AChE inhibitor whose IC_50_ was estimated at 4.09 ± 0.18 µM. The effect of verbascoside on this enzyme activity showed lower linear results than the other two enzymes analyzed (Figure 7). The inhibitory effect of this compound did not reach levels of 50% at the highest tested concentration. Several studies conducted with plant extracts containing verbascoside have shown a powerful inhibition of this enzyme [4]. However, results obtained in this study did not agree with those results. In this sense, results from complex extracts where different compounds could act synergistically together with verbascoside in AChE inhibition could not be compared with those from isolated acteoside. Despite this, verbascoside can produce significant inhibition of the enzyme from the lowest concentrations tested, and therefore it could prove to be of interest as a therapeutic agent in the treatment of Alzheimer’s. According to the cholinergic hypothesis, memory loss in patients with senile dementia or Alzheimer’s is due to a deficiency in cerebral cholinergic function [64]. The use of cholinomimetics and AChE inhibitors are the main therapeutic strategies.

## 4. Conclusions

Verbascoside is a molecule with therapeutic potential in the prevention and treatment of neurodegenerative diseases, especially through its ability to inhibit MAO-A and thereby decrease the degradation of monoaminergic neurotransmitters. This ability, along with its ability to capture free radicals and decrease oxidative stress induced in neuronal cells, demonstrates the interest in this compound in the field of therapeutics, and justifies future studies towards a greater understanding of its mechanism of action.

## Figures and Tables

**Figure 1 antioxidants-09-01207-f001:**
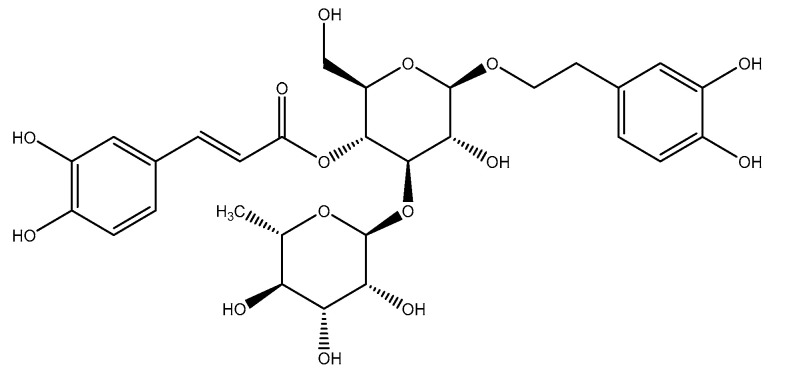
Chemical structure of verbascoside.

**Figure 2 antioxidants-09-01207-f002:**
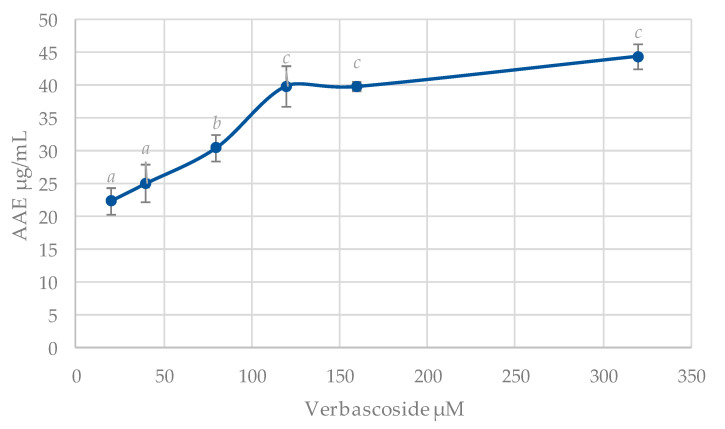
Total antioxidant capacity of verbascoside as ascorbic acid equivalents (AAE µg/mL). The results are expressed as mean ± S.E. of *n* = 3. Different letters indicate statistically significant differences between treatments (ANOVA Bonferroni, *p* < 0.05).

**Figure 3 antioxidants-09-01207-f003:**
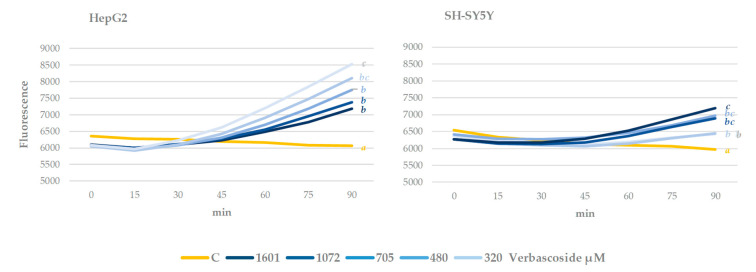
Effect of different concentrations of verbascoside on HepG2 and SH-SY5Y intracellular ROS levels during 90 min of treatment. Cells growing under normal growth conditions. Different letters indicate statistically significant differences between treatments. ANOVA Bonferroni, *p* < 0.05.

**Figure 4 antioxidants-09-01207-f004:**
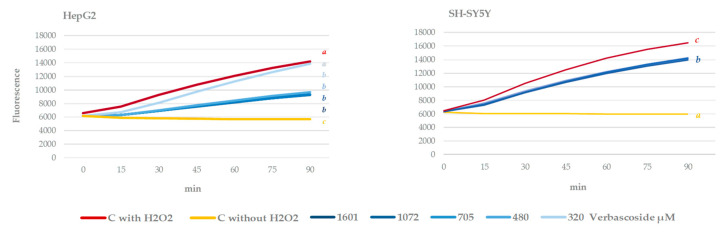
Effect of different concentrations of verbascoside on stressed HepG2 and SH-SY5Y intracellular ROS levels during 90 min of treatment. Cells growing under oxidative stress induced with H_2_O_2_ (200 µM for HepG2, and 500 µM for SH-SY5Y). Different letters indicate statistically significant differences between treatments. ANOVA Bonferroni, *p* < 0.05.

**Figure 5 antioxidants-09-01207-f005:**
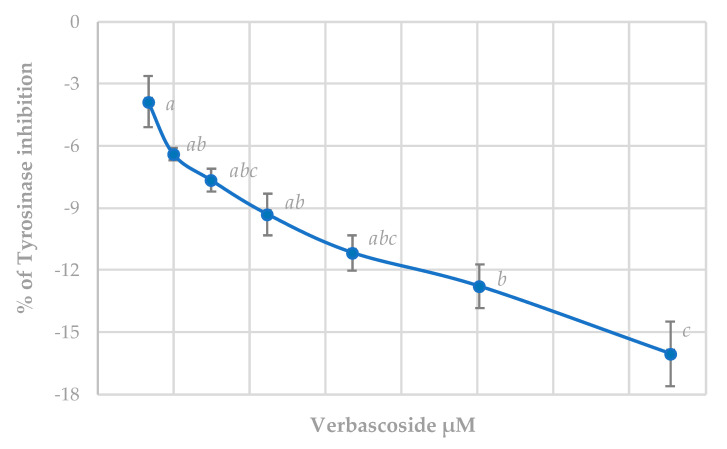
Effect of verbascoside on tyrosinase activity. Results appear as a percentage of inhibition ± standard deviation. Different letters indicate significant differences between treatments (ANOVA Bonferroni *p* < 0.05).

**Figure 6 antioxidants-09-01207-f006:**
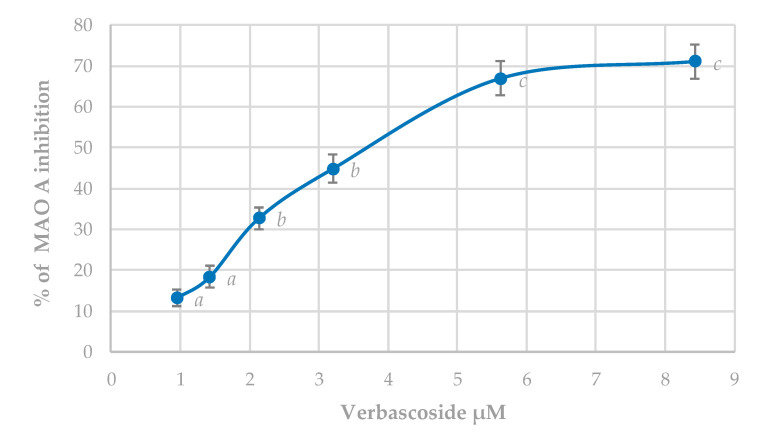
Verbascoside effect on MAO-A enzyme activity. Results appear as a percentage of inhibition ± standard deviation. Different letters indicate significant differences between treatments (ANOVA Bonferroni *p* < 0.05).

**Figure 7 antioxidants-09-01207-f007:**
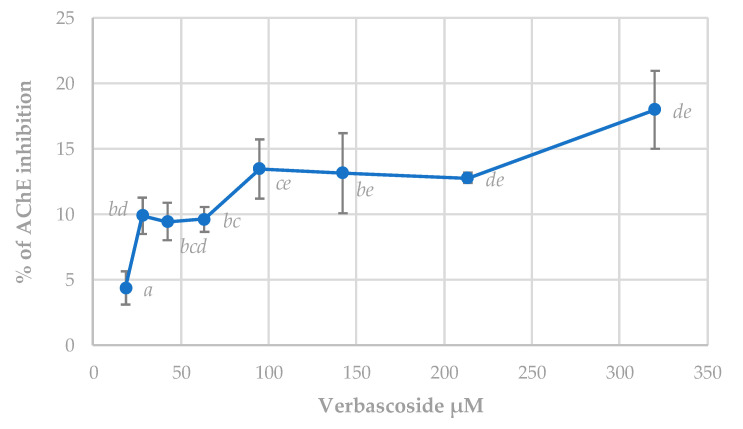
Verbascoside effect on AChE enzyme activity. Results appear as a percentage of inhibition ± standard deviation. Different letters indicate significant differences between treatments (ANOVA Bonferroni *p* < 0.05).

**Table 1 antioxidants-09-01207-t001:** In vitro scavenging activity IC_50_ of verbascoside and ascorbic acid (reference substance) (µM). The results are expressed as mean ± S.E. of *n* = 3.

	DPPH^•^	Superoxide (O_2_^^•^^^−^)	Hydroxyl (^^•^^OH)
Verbascoside	58.1 µM ± 0.6	24.4 µM ± 1.4	357 µM ± 16.8
Ascorbic acid	284.9 µM ± 1.2	66.1 µM ± 1.8	1031 µM ± 19.9

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
