# Peer review of "Neuroprotective Potential of Verbascoside Isolated from Acanthus mollis L. Leaves through Its Enzymatic Inhibition and Free Radical Scavenging Ability"

_antioxidants, 2020, doi:10.3390/antiox9121207_

Round 1

Reviewer 1 Report

The purpose of this study is to isolate and identify verbascoside from Acanthus mollis leaves. The antioxidant capacity of verbascoside was evaluated in vitro through total antioxidant capacity, DPPH•, OH•, and O2•- scavenging activity assays. The effect of verbascoside on intracellular ROS levels of HepG2 and SH-SY5Y cell lines was studied in normal culture and under induced oxidative stress. The authors concluded that the effect of verbascoside on ROS production suggests its potential in the prevention of harmful cell redox changes and in boosting neuroprotection.

Comments:

The authors must explain better the aim. I believe that the main theme of this research was to study the effects of Verbascoside, extracted from Acanthus mollis leaves, in vitro through its antioxidant capacity assessment. On the contrary the authors report “The purpose of this study was to isolate verbascoside from A. mollis leaves.”

The study is interesting, even if some points are unclear. Would be notable go better into dosage/effects analysis.

Point:

Paragraph 1: Introduction is well described, but is too long and some references (i.e line 63-67) are missing. Please provide.

Paragraph 2: Materials and Methods.

This paragraph needs to be revised as regards lines: 187-195 the composition of samples is redundant. Please adjust it.

Maybe it makes more sense move paragraph 2.3 and 2.4 before 2.2 (cell culture), in order to close the description about chemical procedure, extract preparation, and isolation.

Paragraph 2.7: Statistical Analysis. This paragraph must absolutely be revised.

Paragraph 3: Results.

Referring to Table 1, explain better. Verbascoside shows better antioxidant capacity with DPPH respect to Superoxide (O2 •- ) and Hydroxyl (OH•). Why is it happened?

Figures:

Please indicate clearly the significant differences. What means a?, b?, c?

This should be written in the Paragraph 2.7: Statistical Analysis too.

Figure 2: attention in X axis -->0.7 ug/mL is correct?

Figure 4 e 5: keep same scale of Fluorescence for HepG2 and SH-SY5Y i.e. (5000-9000) Fig. 4 and (4000-18000) Fig. 5 respectively.

Figure 6, 7 and 8: how the concentrations and the increases of Verbascoside were chosen? it is not clear!

The impression is of a linear trend ... but it is not true!! The plots must be drown in X axis with the Verbascoside concentration (uM) correctly distanced, otherwise the reader will be confounded.

Acetylcholinesterase (AChE) inhibition results obtained in this study don’t agree with others reported previously. How do you may explain that??

Finally, respect other extracts reported in literature: which is the advantage and the novelty of your verbascoside extract from Acanthus mollis leaves?

Minor point:

line 38: suffers

line 149: CO2

line 248: H2O2

line 441: I suppose figure 6, not 5. Correct?

Reviewer 2 Report

Dear Editors,

This manuscript (MS), antioxidants-988413, “Neuroprotective potential of verbascoside isolated from Acanthus mollis L. leaves through its enzymatic inhibition and free radical scavenging ability” is the observation that analyze the neuroprotective effect of verbascoside in vitro and cell cultures. It's worth noting that verbascoside exerts neuroprotective activity in HepG2 and SH-SY5 cells. This finding might be useful for the development of neuroprotective medicine. However, I recommend that this MS not be accepted without some revision. This MS is very difficult for the reader to understand. For example, the authors describe that Line 341-345 is the explanation in Fig. 2, but it is actually the explanation in Table 1. Furthermore, there is no explanation for Fig. 6. Also, there are many typographical errors.

The main data of this manuscript is that verbascoside has neuroprotective potential in SH-5Y5Y. However, other researcher has already been reported that nutraceutial product, rich in verbascoside, neuroprotective potential in SH-SY5Y cells [PMID; 32882797]. Thus, it can be judged that the novelty of this MS is low.

It is also worrisome that the concentration of verbascoside used differs depending on the assay. Plant-derived compounds have a wide variety of effects at high concentrations. If these concentrations deviate from the physiological concentration range, this MS's data can be an artificial effect.

Reviewer 3 Report

A very interresting paper with a great number of experimental conditions. It appears that the discussion on the  effect on purified enzymes in comparison with the effect on cells is very difficult because in one case you have the direct effect and in the other case you have the direct AND indirect effect… for example verbascoside is reported to be an inhibitor of protein kinase C ( Will it be reasonable to discuss that point in one place or another in the article ??…) so the final effect on cells and enzymatic activities ( for example  ROS, tyrosinase…) must be considered and discussed cautiously even if the duration of contact is small..

Concentrations in µM  is better than mg or µg in order to discuss the final results and claim differences. Please, if possible, use an homogeneous expression of concentrations throughout the article… Finally, what is the final composition of your extract ? is it a pure verbascoside solution ?

Figure 2 what is the number of experiments ? n=3 is the number of points in each experiments ? have you done 3 experiments with 3 points ? I think that a conclusion is not possible  due to the  low number of experiment

Figure 3 I don’t understant the statistical analysis a c ??? what does it means ?

Line 389 : « Regarding HepG2, an increase in the clearly dose-dependent manner of the ROS levels was observed, which was significantly higher in cells treated with the lowest concentration of verbascoside, and less evident with the higher concentrations tested ». Sorry I don’t understand this sentence… a clear dose-dependent increase is less evident at the highest dose ?? why don’t you test lower concentrations ? (mg/ml on cells and µg/ml in acellular texts ??)

Line 441 presentation is confusing…  a negative inhibition is quite acceptable but the reader is sometimes lost…  perhaps the presentation of an increase of activity may be more simple! statistical analysis must be also explained.

Line 479 Several studies conducted with plant extracts containing verbascoside have shown a powerful inhibition of this enzyme [67]. As far as I know the cited publication is not very accurate about the composition of the tested product...I suggest a comment on the fact that the activity of a solution with single product must be cautionously extrapolated to a solution containing the same compound in combination with numerous other compounds with synergy / antagonism of action…

Reviewer 4 Report

The paper entitled „Neuroprotective Potential of Verbascoside Isolated from Acanthus mollis L. Leaves through its Enzymatic Inhibition and Free Radical Scavenging Ability” aims to isolate verbascoside from A. mollis leaves and analysis towards neuroprotective and free radical scavenging capacity.

The presented studies were well planned and methodology is appropriate for the aim of study.  Authors decided to use various mathods. In order to determine antioxidant activity and against free radicals the following methods used: DPPH, xanthine/xanthine oxidase assay, hydroxyl radical scavenging capacity, total antioxidant capacity (TAC), cell culture antioxidant capacity. In order to determine antineurodegenerative activity the following assays were performed: tyrosinase inhibition assay, MAO-A inhibition assay, and acetylcholinesterase (AChE) inhibition assay. Additionally statistical analysis of the studies was performed.

In my opinion the manuscript is well prepared and obtained results are interesting and novelty. The results are well analyzed and form of their presentation is good.  References are adequate. Discussion is comprehensive and readable. Authors decided to focus on one of the most important problem, namely neurodegeneration and disorders link with the changes in whole organism, especially in brain. Authors decided to use evaluate activity of compound isolated from plant commonly used in Asia and Pacific region. It is good idea due to the plant can be used as functional food having additional advantages, especially for the elderly but simultaneously for prevention in younger people. The obtained results are interesting and can be interpreted as good direction for future studies.

In my opinion the paper can be published in Antioxidants.

Round 2

Reviewer 1 Report

Dear Authors, 

I think that:

-the introduction is yet too long

-because figure 2 is not represented as figure 5, 5, 7??it is better to standardize

By the ways, all other points have been carried out.

Author Response

Thank you for the arduous work to try to improve the article.  We believe that the paper has been significantly improved after introducing your suggestions.

Introduction is now shorter.

Figure two has been changed and now is represented as 5, 6 and 7.

Kind Regards

Authors

Reviewer 3 Report

thank you for the modifications of your article, it appears acceptable for publication

Author Response

Thank you for your arduous work improving the article. 

Kind Regards

Authors
